# Postoperative Chylothorax in Neonates and Infants after Congenital Heart Disease Surgery—Current Aspects in Diagnosis and Treatment

**DOI:** 10.3390/nu14091803

**Published:** 2022-04-26

**Authors:** Georgios Samanidis, Georgios Kourelis, Stavroula Bounta, Meletios Kanakis

**Affiliations:** 1Department of Adult Cardiac Surgery, Onassis Cardiac Surgery Center, 17674 Athens, Greece; 2Department of Pediatric and Adult Congenital Heart Surgery, Onassis Cardiac Surgery Center, 17674 Athens, Greece; meletis_kanakis@yahoo.gr; 3Pediatric Cardiac and Adult Congenital Heart Disease Intensive Care Unit, Onassis Cardiac Surgery Center, 17674 Athens, Greece; gkourelis@yahoo.gr (G.K.); linabouda@hotmail.com (S.B.)

**Keywords:** chylothorax, congenital heart disease, breast milk, MCT, low fat diet, nutrition

## Abstract

Postoperative chylothorax is a rare complication following cardiac surgery for congenital heart disease (CHD) in the pediatric population, including neonates and infants. Multiple mechanisms are involved in the development of postoperative chylous effusions, mainly the traumatic injury of lymphatic vessels, systemic venous obstruction and dysfunction of the right ventricle. In this review, we focus on the existing evidence regarding the definition and diagnosis of postoperative chylothorax in children with CHD, as well as current therapeutic approaches, both nutritional and interventional, for the management of these patients. As part of nutritional management, we specifically comment on the use of defatted human milk and its effect on both chylothorax resolution and patient growth. A consensus with regard to several key aspects of this potentially significant complication is warranted given its impact on the cost, morbidity and mortality of children with CHD.

## 1. Introduction

### 1.1. Definition and Epidemiology

Chylothorax refers to the accumulation of lymphatic fluid in the pleural space. Its presence in the neonatal and pediatric population has been associated with numerous conditions, mainly chest trauma, thoracic surgery, extracorporeal membrane oxygenation and primary or metastatic malignancy, particularly lymphoma. In newborns rapidly increased venous pressure during delivery may lead to thoracic duct rupture. Less common causes include thrombosis of the duct, superior vena cava or subclavian vein, lymphangiomatosis, restrictive pulmonary diseases, tuberculosis, histoplasmosis and congenital anomalies of the lymphatic system [1,2].

Postoperative chylothorax is a known complication after pediatric cardiac surgery for congenital heart disease (CHD), and its incidence increased consistently over the last decades, a finding that could be reflecting the increased complexity of surgical procedures performed. It has been speculated though that augmented awareness of this complication has led to increased diagnosis rather than a true increase in incidence [3,4,5]. According to reports from the Pediatric Cardiac Critical Care Consortium (PC4) and Pediatric Health Information System (PHIS) databases [5,6], the overall incidence of chylothorax in pediatric patients following congenital heart surgery or heart transplantation ranges between 2.8–3.8%, with higher rates being observed in neonates (6.9%) and patients with single ventricle physiology (6.9%), chromosomal/genetic anomalies (5.2%) and major noncardiac anomalies (6.4%). 

Thrombotic events related to upper extremity central venous catheters (CVC) (OR 10.63; 95% CI 4.24–26.64 when compared with no upper extremity CVC group) and neonatal population (OR 4.7; 95% CI 2.7–8.1 when compared with children > 1 year old) yielded the highest OR in a multivariate adjusted model evaluating risk factors for chylothorax development after cardiac surgery in children [6].

Several mechanisms are implicated in the development of postoperative chylothorax after CHD surgery, such as damage to the thoracic duct or collateral lymphatics during dissection or superior vena cava cannulation and increased systemic venous pressures associated with cavopulmonary anastomosis (Glenn shunt or Fontan operation) or right ventricular dysfunction. Further risk factors include the type and complexity of the surgical procedure performed as well as the duration of both the cardiopulmonary bypass and aortic cross-clamp [7,8]. 

With respect to the type of surgery, atriopulmonary and cavopulmonary anastomoses have been associated with the highest incidence of postoperative chylothorax (5.7%), followed by operations related to the complete correction of transposed great arteries (4.3%), the repair of aortic arch anomalies (3.7%), such as coarctation of the aorta and interrupted aortic arch and the complete repair of total anomalous pulmonary venous drainage (3.7%), while repair of atrial septal defect is associated with lower incidence of chylothorax (0.9%) [5].

### 1.2. Purpose of Review

The scope of this narrative review is to indicate the need to address several key aspects in the definition, diagnosis and management of postoperative chylothorax in neonates and infants after surgery for CHD. Given the lack of consensus in regard to the overall management of these patients, we focus on the available evidence relevant to current therapeutic approaches, with special reference to nutritional and interventional treatment options. In particular, we discuss the use of defatted human milk (DHM) and its impact on both chylothorax resolution and patient growth as part of the nutritional management of this special population, while we comment on current advances on interventional and surgical treatment options for patients who fail to respond to conservative management.

## 2. Materials and Methods

A comprehensive search in the Medline and PubMed databases was conducted using the following search terms: chylothorax, cardiac surgery and congenital heart disease. Additional filters included pediatric studies (patients < 18 years old) published in the last 10 years. With regard to type of studies, we mainly included retrospective observational and case—control studies, as well as reviews. Exclusion criteria were case reports; studies with a mixed patient population (children and adults), unconfirmed diagnosis of chylothorax, or chylothorax unrelated to cardiac surgery. Title and abstract screening were performed by two researchers independently. Outcomes of interest included reported nutritional and interventional approaches in the management of neonates and infants with CHD, who developed postoperative chylothorax after cardiac surgery. Backward citation searching was used to identify relevant studies missed by the initial search.

## 3. Results

A total of 77 articles were identified using the reported search for title and abstract screening. Papers considered to be of importance based on outcomes of interest are presented and commented on.

Rocha et al., in their recent review [2], suggested utilizing chest radiography or ultrasound to classify pleural effusion as small, moderate or large (Table 1). Moreover, they considered a chylothorax with continuous drainage > 10 mL/kg/day to be as high volume. On the other hand, other researchers used a cut-off value of 20 mL/kg/day to discriminate between low and high output chylothorax [9,10]. To add more confusion to the aforementioned discrimination, an output of >5 mL/kg/day for at least 5 days has been used in the literature to define high-output states [11]. 

According to Rocha’s et al. approach for post-traumatic chylothorax, with cardiovascular-thoracic-surgery-related chylothorax being classified as such, asymptomatic patients with small effusion can managed with either bowel rest and parenteral nutrition or nutritional measures. The latter include either MCT diet or modified breast milk, along with 3 g/kg/week of lipids (once weekly or in three divided doses regime) plus fat-soluble vitamins. Octreotide infusion remains an option in all patients. Low-fat diet (either MCT or modified breast milk) can be gradually replaced with adapted formula or breast milk once chylothorax is dry.

On the contrary, medium or large volume effusions in symptomatic patients with respiratory compromise are proposed to be managed with TPN and bowel rest until sparse drainage (<2 mL/kg/day). Subsequently, patients can be commenced on either MCT formula or modified breast milk in small amounts with gradual increase, along with 3 g/kg/week of lipids and fat-soluble vitamins. Additional medical management for high output drain losses, especially when prolonged, includes partial replacement with human albumin and NaCl 0.9% as well as the use of octreotide infusion, monitoring and appropriate correction for immunoglobulin, clotting factor and bicarbonate deficiencies [2].

Once a good response, defined as a total drainage volume < 2 mL/kg/day with enteral only MCT or modified breast milk diet and no medications, is noted, the chest drains can be removed. MCT or modified breast milk diet is suggested to continue for 6 weeks and to be gradually replaced with adapted formula or breast milk. Conservative treatment may continue for 3–4 weeks for partial response, defined as drainage < 10 mL/kg/day, before considering invasive procedures. Invasive approaches are proposed with an output > 10 mL/kg/day after 1 week of conservative treatment, or >100 mL/day for 5 consecutive days or the development of difficult to control metabolic and nutritional complications [2].

Winder et al. proposed a more conservative approach with good results [12]. In particular, all patients diagnosed with chylothorax were initially commenced on medium—chain triglyceride (MCT) feeds or low-fat diet (<10 g/day) for 24–36 h before classifying chest tube output as high (>20 mL/kg/day) or low (≤20 mL/kg/day). Although high-output states were initially managed with nil-per-os (NPO) and parenteral nutrition, there was constant reevaluation regarding output status. If, on any day, the chest tube output was <10 mL/kg/day, patients were reassigned to low output arm and commenced the MCT diet. 

Eventually, the MCT duration was reduced to 4 weeks instead of 6, which is typically reported in the literature [9,13], without an increase in adverse effects. Using the described approach, authors managed to reduce the number of patients in the high output arm from 12 to 8, thus eliminating at least 12 NPO days across these four patients. Moreover, once a low output state was achieved, eight out of nine patients remained in this state after initiation of feeds, regardless of NPO duration, which ranged from 1 to 17 days. Of note, shortening MCT feed duration can have clinical implications, as this diet is deficient in several nutritional factors and associated with poor weight gain [14].

Christofe et al. [15] described utilizing a combination of fasting and TPN (+/− octreotide infusion) in 82% of children who developed chylothorax after surgery for CHD. Although authors concluded that fasting combined with TPN to be the best strategy regarding resolution of chylothorax in <14 days, they do not provide details regarding chylous output, while no defined chylothorax protocol is being described. These reasons along with the retrospective study design pose significant difficulties in drawing safe conclusions.

Few small studies have examined the effectiveness of DHM over MCT formula in the management of newborns and infants with chylothorax development after cardiac surgery for CHD, with regard to chylous effusion resolution and patient growth outcomes. Neumann et al. [16] analyzed 23 patients, 13 of which were treated with DHM and 10 with MCT formula and found no statistically significant difference in the overall volume and the duration of chest tube drainage between the two groups. 

Furthermore, no significant differences were reported regarding the lengths of ICU and hospital stay, duration of tube feeding and growth outcome measures. Patients in the DHM group received a low-fat diet for a significantly shorter time (41 ± 5 vs. 67 ± 58 days respectively, *p* = 0.036). Of note, patients in both groups experienced a decrease in all growth parameters 3 months after the diagnosis of chylothorax, providing some evidence in favor of DHM fortification with MCT oil and milk fortifiers to optimize caloric and nutritional supply.

Fogg at al. [17] retrospectively compared outcomes between 14 infants fed with DHM versus 21 treated with MCT-based formula. DHM showed comparable efficacy with MCT formula regarding chylothorax resolution, while no significant differences in the duration of mechanical ventilation, length of hospital stays, recurrence of chylothorax and mortality between the two groups were reported. Although infants treated with DHM had higher weight-for-age z-scores at hospital discharge, with authors suggesting possible nutritional advantages of DHM over MCT, careful interpretation of this finding is warranted, as infants in the MCT group were smaller at birth (*p* = 0.06 for weight-for-age z-score), and there was no significant difference in the absolute growth velocity from admission to discharge between the two groups.

Kocel et al. [14] compared outcomes between eight infants treated with DHM and eight infants treated with MCT-based formula. Although both nutritional strategies were equally effective in the treatment of chylothorax, authors highlighted suboptimal growth in either group. Moreover, they reported some evidence of slower growth in the DHM group, including a statistically significant decline from enrollment to study completion in weight-for-age and length-for-age z-scores, emphasizing the need for the fortification of DHM.

DiLauro et al. [18] in a recent randomized control trial evaluating modified fat breast milk (MFBM) feeding protocols specifically designed to support growth in infants with chylothorax after cardiac surgery, found no differences in chylous effusion resolution and growth parameters between patients receiving fortified MFBM (two groups consisting of eight patients each; each group was treated with a specific proactive MFBM protocol) and MCT formula (one group consisting of eight patients).

Poor growth in children with CHD has been described in numerous studies [19,20]. Fortification of DHM is considered to be required according to most studies examining its growth effects on children who develop chylous effusions after cardiac surgery, in order to provide adequate calorie and micronutrient intake, as it has been found to be deficient in calories, proteins, essential fatty acids and fat-soluble vitamins [16,17,21,22,23]. 

Jackson et al. [22] additionally indicated a slightly increased carbohydrate content of 0.2 g/100 mL in DHM compared with full-fat human milk (FFHM), a finding that is unlikely to be of clinical importance. They highlighted though the need to carefully monitor infants who are treated with DHM because of the potential to receive high carbohydrate concentration, which, in the long run, can have significant consequences in their growth and metabolism, as it stimulates lipogenesis and fat deposition while increasing the respiratory quotient. In contrast to these reports, Barbas et al. [24] reported similar mean protein contents between defatted and unprocessed human milk.

The immunologic properties of DHM in comparison with FFHM were studied by Jackson et al. [22]. Despite having fewer immune cell populations, including lymphocytes, monocytes and neutrophils, DHM and FFHM were reported to contain comparable levels of lactoferrin, lysozyme and both immunoglobulins A and G. Moreover, bacterial growth inhibition against Streptococcus pneumoniae and Escherichia Coli was equivalent between DHM and FFHM. Although reduced, the presence of immune cells on DHM possibly offers significant advantages over MCT-based formulas. In opposition to these findings, Drewniak et al. [23] reported lower levels of IgA in DHM, especially when separation was performed by centrifugation, regardless the temperature, rather than refrigeration.

## 4. Discussion

### 4.1. Diagnosis of Postoperative Chylothorax

Diagnosis of chylothorax is usually based on pleural fluid analysis, which is typically characterized by a white blood cell count > 1000 cells/μL with >70–80% lymphocytes and, in enterally fed patients, a triglyceride concentration > 110 mg/dL. A pleural fluid triglyceride level greater than serum triglyceride level is also characteristic of chylous effusions. Chylous fluid is classically being reported as milky or turbid. Nevertheless, this can be misleading, especially in fasted patients. 

Chylothorax in the postoperative period is usually suspected and thus should be checked for when chest tube drainage either becomes persistent or develops the characteristic milky appearance, especially after the patient has been fed, as well as when a similar appearing fluid is found after draining a newly discovered effusion [10,13,25,26]. Of note, we need to mention that there are no specified diagnostic criteria that need to be met before applying a therapeutic protocol for chylothorax management [2,3,4,12]. Qualitative methods for diagnosing chylothorax, such as the microscopic examination of fluid stained with Sudan III showing fat globules, have been described in the literature as well, adding further complexity and confusion with regard to chylothorax definition [27].

For diagnostic purposes, we advocate the use of an algorithm, such the one proposed by Skouras et al. [28], which is based on pleural fluid analysis and, if needed, lipoprotein electrophoresis. Pleural fluid triglyceride levels > 110 mg/dL or <50 mg/dL establish or exclude the diagnosis of chylothorax, respectively, for most instances [28], while the presence of chylomicrons in pleural fluid as demonstrated by lipoprotein electrophoresis, which is considered to be the gold standard in the diagnosis of chylothorax [29], clarifies the field in ambiguous cases, such as those with pleural fluid triglyceride levels 50–110 mg/dL or those involving fasted (or malnourished) patients. The proposed algorithm for the diagnosis of postoperative chylothorax is presented by Skouras et al [28].

### 4.2. Treatment Goals and Current Therapeutic Approaches in the Management of Postoperative Chylothorax in Children after Surgery for CHD

Chyle constitutes of cellular components with lymphocytic predominance, lipids, electrolytes, glucose, bicarbonate and proteins, such as albumin, clotting factors, complementary elements, antibodies and peptide hormones. Excessive and prolonged chylous losses can result in homeostatic disorders, protein energy malnutrition, clotting and immunological abnormalities, rendering these patients susceptible to significant risks, including, but not limited to, hypovolemia, hypoproteinemia, thrombi formation and systematic infections [2,3]. Chylothorax after pediatric cardiac surgery has been associated with an increased duration of postoperative mechanical ventilation, intensive care unit (ICU) and hospital lengths of stay, cost and mortality [5,6].

The main treatment goals for chylous effusions are to decrease and, eventually, cease the thoracic lymph flow and allow for the lymphatic vessels to heal or develop [2,3]. Conservative nutritional management strategies include either a combination of nil-per-os (NPO) and total parenteral nutrition (TPN), or enteral feeding with MCT diet, fat-modified breast milk or low-fat diet. While conventional fats are absorbed via lymphatic system and require chylomicron formation, medium-chain triglycerides (MCT) are absorbed by intestinal cells and directly transported into the portal system without chylomicron formation [30]. An MCT-based diet, from a pathophysiologic perspective, potentially could reduce lymphatic flow along the thoracic duct and, thus, minimize chyle output drainage. 

Pharmacological agents, such as octreotide, can be added as adjunctive therapy to reduce the lymphatic flow. Additional medical management including, but not limited to, respiratory support, the optimization of cardiovascular performance, immunoglobulin and clotting factor supplementation and correction for electrolyte and bicarbonate deficiencies, is offered mainly on a case-by-case basis guided by institutional or expert opinions, as evidence-based treatment choices for post-operative chylothorax in children and older children are lacking [2,4].

Despite generally accepted medical and, under certain circumstances, interventional or surgical management for postoperative chylothorax in neonates and infants following cardiovascular surgery, there is a lack of consensus regarding both a definition that would lead to a change in patient management and the output classification of chylous effusions as low and high as well as the optimum care of these patient population [4]. Nutritional support management strategies vary depending on center and physician experience and/or preference. 

While most studies suggest enteral nutrition with MCT for low drainage output chylothorax after cardiovascular surgery in children [3,4,9,27,31], significant variance of practice exists regarding nutritional management of high-output chylous effusions. Several authors suggest NPO with bowel rest and TPN initially [2,9,12,27], while others propose enteral feeding with MCT [3,4,31,32]. Of note, there is significant inconsistency among studies regarding the duration of enteral feeding in high-output states. 

Most authors agree though that it is reasonable to switch from an MCT enteral diet to TPN for unresolved high output chylous drainage (i.e., >10 mL/kg/day), paying special attention to excluding raised central venous pressures or central venous thrombosis. Clinicians should not underestimate the potential complications related to TPN administration, such as venous thrombosis and sepsis, as well as the protective function of enteral feeding on the gut barrier through luminal nutrient provision [3,4,31].

Octreotide (or somatostatin) can be added as a pharmacologic adjunct in the management of young infants with postoperative chylothorax. Its use has been described in several published institutional protocols, with most authors suggesting its utilization as a second line treatment [2,3,12,15,33,34]. Mixed data regarding octreotide’s effect on hospital length of stay and chylothorax duration have been reported in the literature, which could be attributed, at least partially, to the differences of local protocols, namely the time interval between the onset of chylothorax and the commencement of octreotide infusion [35]. Nevertheless, although the use of octreotide is mainly based on reports from children with congenital chylothorax, representing a different entity from postoperative chyle leak [4], it can be useful in reducing the volume of chylous effusions postoperatively after failure of conservative approach, with its use accompanied by an acceptable safety profile [33,36].

### 4.3. The Use of Defatted Human Milk (DHM) as Part of the Nutritional Management in Neonates and Infants with Cardiac Surgery Related Chylothorax

Breast milk is considered to be the perfect nutrition for infants. Despite that, it is typically excluded from the feeding regime of infants with chylothorax due to its high long-chain triglyceride (LCT) content [37]. MCT-based formulas are used instead, as discussed previously, based on their properties to reduce lymphatic flow [31,38,39]. However, these infants may well benefit from human milk over cow’s milk-based formulas or other substitutes due to the unique composition and favorable properties, such as the well-described protective effects over childhood leukemia and late metabolic diseases, particularly obesity and type 2 diabetes, as well as the beneficial impacts on the immune system, intestinal function and brain development, to name a few [38,39,40].

Several human milk fat separating methods for DHM preparation have been described in the literature, including centrifugate (refrigerated or room temperature), portable cream separator and refrigeration methods. Centrifugate and cream separator methods are reported to be more effective in removing fat. After separation, a spatula, spoon or syringe can be used to remove the fat layer and to pour the fat-free liquid into collection cups for further use [16,17,21,22,23,24]. A detailed analysis of each method is beyond the scope of this review. 

Of note, although LCT contributes to 13–17% of the fat content (or 4–7% of the energy content) in commercial MCT-based formulas, an optimal LCT content in DHM has not been established [24]. Fogg et al. [17] accepted a caloric contribution from fat < 3% to be adequate in DHM, while other researchers used a creamatocrit ≤ 1% as a cut-off value [14]. The creamatocrit method for assessing residual fat, although accurate for unprocessed human milk, should be used with caution with DHM, as it can significantly overestimate the fat and energy content [24]. 

### 4.4. Interventional or Surgical Management of Chylothorax after Congenital Heart Surgery

Interventional or surgical treatment remains an option for patients that fail to respond to conservative management as addressed above. Guided by clinical criteria, such as respiratory compromise, haemodynamic status and metabolic disorders and in order to reduce imminent morbidity and mortality, surgical management is generally indicated when the chylous output exceeds 100 mL/kg/day (or 100 mL/year of age) for five days, 100 mL/day persistently for >2 weeks, or remains unchanged for 1–2 weeks, despite conservative management [7,41]. Clinical improvement while on conservative management should postpone the invasive treatment, otherwise surgery or interventional management should be employed.

In cases of thoracic duct injury, identification and ligation of the leakage site or the thoracic duct can be accomplished by administrating cream or oil via a nasogastric tube intraoperatively. If the leakage point or duct cannot be identified, massive ligation of the duct along with adjacent tissue should be performed. Thoracic duct ligation is reported to be successful in 90% of cases; nevertheless, this procedure is associated per se with mortality, which can be high as 20% [34,41]. 

Right thoracotomy or VATS is the surgical treatment of choice for bilateral chylothorax, while a pleuro-pericardial window is necessary in the case of pericardial chyle effusion. In infants, thoracoscopic parietal pleural clipping is feasible and associated with safe and successful results [42]. From our experience, in rare cases where massive chylothorax is present due to severe postoperative ventricular dysfunction secondary to residual lesions with significant hemodynamic compromise, a reoperation addressing the responsible technical aspects of the prior operation should be reserved as the definite treatment.

Pleurodesis, a chemical procedure involving obliteration of pleural space using various chemical substances including autologous blood, has also been employed in cases of refractory postoperative chylothorax. Takahashi et al. [41] proposed pleurodesis using OK432 as a surgical first-line therapy for chylothorax even for neonates. The authors indicated the necessity to apply pleurodesis for refractory postoperative chylothorax at least within one month, when conservative therapy proves to be ineffective.

Scarce data from literature indicate successful thoracic duct embolization with micro-catheters using lipiodol, coils, or glue in infants who underwent congenital heart surgery. However, percutaneous thoracic duct embolization has been associated with several complications, such as stroke, pulmonary oedema and lymphedema [7,43,44]. Some investigators perform lymphovenous anastomosis in pediatric patients suffering from chylothorax secondary to lymphatic injuries, who have either failed conventional medical management or are not candidates for alternative interventions. This procedure is usually a multidisciplinary collaboration among surgeons, interventional radiologists and lymphedema specialists [45].

### 4.5. Suggestions for Further Research

In addition to the proposed diagnostic approach, we consider the topics summarized in Table 2 to be of significant importance for a universal protocol regarding the management of postoperative chylothorax in neonates and infants after cardiac surgery for CHD to be developed. As discussed above, further investigation is warranted regarding these issues, to thus achieve the best evidence-based practice.

## 5. Conclusions

Chylothorax, although relatively rare, is a potentially serious complication for neonates and infants undergoing surgery for CHD and is associated with increased cost, morbidity and mortality. Early diagnosis and treatment are necessary to mitigate its numerous adverse effects. An overall consensus regarding optimal management is needed.

Several key questions need to be answered, such as the chylothorax diagnostic process (biochemical criteria and qualitative methods), number of criteria that need to be met before triggering the management protocol, classification of chylous drainage as high or low output, indications for bowel rest and TPN, MCT versus DHM when enteral nutrition is indicated, duration of conservative management before proceeding with invasive treatment and indications for surgical or interventional management. In our review, we offer an algorithm for postoperative chylothorax diagnosis based on the available evidence, while we propose several topics that need to be further investigated. Large multicenter studies are needed to define guidelines for the management of these patients.

Since the clinical efficacy of TPN over low LCT enteral feeding has not been established and until more evidence become available, we advocate low fat enteral feeding over bowel rest and TPN, as it is easier to apply, while providing gut barrier protection and avoiding potential complications related to TPN administration. Moreover, we recommend DHM versus MCT formulas as the standard of care, so that neonates and infants can benefit from the numerous well-known advantages of human milk. 

Portable cream separator methods appear to be user-friendly and reliable in removing fat from human milk even after hospital discharge. Fortification is needed to prevent growth failure and, ideally, should be tailored to each individual’s needs. Using the creamatocrit method to estimate the fat and caloric content of DHM may overestimate the fat content, thus, leading to inadequate fortification. Evidence-based separating methods with optimal levels of LCT and other macronutrients in DHM are needed while paying attention to retaining the full immune cell population.

## Figures and Tables

**Table 1 nutrients-14-01803-t001:** Definitions of the volume of chylothorax effusion.

	Chest Radiograph	Chest Ultrasound ^a^
**Small**	<¼ of the lung field	<10 mm
**Moderate**	¼–½ of the lung field	10–30 mm
**Large**	>½ of the lung field	>30 mm

^a^ chest ultrasound was performed at the posterior pleural costophrenic recess.

**Table 2 nutrients-14-01803-t002:** Suggestions for further research.

Chylothorax classification as low and high output
Indications for bowel rest and total parenteral nutrition
Duration of conservative management before classifying a patient as non-responder
Relative and absolute indications for interventional and surgical treatment
Role and indications of adjunctive therapies such as octreotide
Development of standardized separating and fat removal methods for human milk
Defatted human milk fortification individualized to patient needs

## Data Availability

Not applicable.

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
