# Peer review of "Postoperative Chylothorax in Neonates and Infants after Congenital Heart Disease Surgery—Current Aspects in Diagnosis and Treatment"

_nutrients, 2022, doi:10.3390/nu14091803_

Round 1

Reviewer 1 Report

In their manuscript, the Authors focused on a rare but severe complication following cardiac surgery for congenital heart disease (CHD), namely postoperative chylothorax. To date, international consensus treatment guidelines for diagnosis and management of this condition are lacking. Therefore, this paper could be a useful update about this relevant topic.

I believe that the following points should be addressed.

  1. Introduction

The Authors aim to focus on chylothorax in the specific setting of congenital cardiac surgery (as they mention in the title of the article); therefore, they should include a list of the incidence of postoperative chylothorax based on the type of surgery (also in a table).

In the paragraph about Diagnosis and treatment goals, the Authors should mention: 1) the correct time to test the pleural fluid for chylothorax after cardiac surgery; 2) the differential diagnosis with pseudochylothorax.

  1. Discussion

The recent paper by Ha Lee, et al. (Postoperative Chylothorax after Congenital Cardiac Surgery,  Korean J Thorac Cardiovasc Surg. 2020;53(2):41-48) should be mentioned and discussed. Moreover, the role of octreotide should be examined and discussed (Aljazairi AS, Bhuiyan TA, Alwadai AH, Almehizia RA. Octreotide use in post-cardiac surgery chylothorax: a 12-year perspective. Asian Cardiovasc Thorac Ann. 2017 Jan;25(1):6-12. doi: 10.1177/0218492316682670. Epub 2016 Dec 11. PMID: 27920229. Sudharsan Madhavan, Masakazu Nakao, How efficacious are Octreotide and Somatostatin in the management of chylothorax in congenital cardiac surgical patients?, Interactive CardioVascular and Thoracic Surgery, Volume 33, Issue 5, November 2021, Pages 773–778, https://doi.org/10.1093/icvts/ivab155)

A summary table with a proposal for diagnosis and treatment of postoperative chylothorax should be inserted.

Author Response

REVIEWER 1

Comment #1:

  1. Introduction

The Authors aim to focus on chylothorax in the specific setting of congenital cardiac surgery (as they mention in the title of the article); therefore, they should include a list of the incidence of postoperative chylothorax based on the type of surgery (also in a table).

ANSWER:

Both text (lines 57-62) and a table (table 1) have been added in the introduction section.

Comment #2:

In the paragraph about Diagnosis and treatment goals, the Authors should mention: 1) the correct time to test the pleural fluid for chylothorax after cardiac surgery; 2) the differential diagnosis with pseudochylothorax.

ANSWER:

  • Correct timing for testing has been added in section 4.1 (lines 246-250)
  • Pseudochylothorax is not usually in the differential diagnosis for postoperative pleural effusion development in neonates and infants after cardiac surgery for CHD. Cholesterol effusions are long-standing pleural effusions, more frequently found in middle-aged men, with an average latency period of 5 years and a range of 11 to 15 years. Tuberculosis, rheumatoid arthritis, empyema and malignancies are reported to be the most common diseases associated with cholesterol effusions (Agrawal V, Sahn SA. Lipid pleural effusions. Am J Med Sci. 2008;335(1):16-20). We have added a couple of references (ref no 27, 28) for the readers, but we don’t feel we should add any further comments as this is not the case in the studied population.

Comment #3:

Discussion

The recent paper by Ha Lee, et al. (Postoperative Chylothorax after Congenital Cardiac Surgery,  Korean J Thorac Cardiovasc Surg. 2020;53(2):41-48) should be mentioned and discussed. Moreover, the role of octreotide should be examined and discussed (Aljazairi AS, Bhuiyan TA, Alwadai AH, Almehizia RA. Octreotide use in post-cardiac surgery chylothorax: a 12-year perspective. Asian Cardiovasc Thorac Ann. 2017 Jan;25(1):6-12. doi: 10.1177/0218492316682670. Epub 2016 Dec 11. PMID: 27920229. Sudharsan Madhavan, Masakazu Nakao, How efficacious are Octreotide and Somatostatin in the management of chylothorax in congenital cardiac surgical patients?, Interactive CardioVascular and Thoracic Surgery, Volume 33, Issue 5, November 2021, Pages 773–778, https://doi.org/10.1093/icvts/ivab155)

Answer:

  • The paper by Ha Lee, et al., although quite interesting, cannot be included in our review since it includes a mixed population of children and adults, without reporting results separately for children. We acknowledge though, that authors describe a clear management protocol for postoperative chylothorax in the pediatric population, thus we have added a reference (ref. no 14).
  • Our review mainly focuses on the nutritional rather on the pharmaceutical element of conservative management of postoperative chylothorax. We have added a paragraph on octreotide as per suggestion (section 4.2 lines 315-326), as we acknowledge that it can be a useful and safe adjunct to conservative management of these patients. Further expanding on that topic would be out of scope for this review.

Comment #4:

A summary table with a proposal for diagnosis and treatment of postoperative chylothorax should be inserted.

ANSWER:

We have added our proposed algorithm for the diagnosis of postoperative chylothorax (section 4.1 lines 256-264) plus a diagram (figure 1). Regarding treatment we cannot really provide an evidence-based suggestion, as this is one of the main problems pointed out in our study: further research is needed to determine the best approach. In our center we follow a treatment protocol based on our experience, but we cannot recommend a general guideline because there are quite a few questions that need to be addressed, as highlighted in our review. We have added though a table with our suggestions for further research in the topic at the end of the text (table 4).

Reviewer 2 Report

Dear authors,

This is an interesting review. Please allow me some suggestions:

  • Title: ...congenital heart disease surgery 
  • A subchapter explaining the purpose of the study would be welcome 
  • The Material and Methods chapter is missing, and also the Results section. How was this review done? How many articles were analyzed, what were the inclusion / exclusion criteria? What databases were consulted?
  • Your first sentence in the Conclusions chapter talks about " increased cost, morbidity and mortality", but you have not previously discussed all these issues
  • Your review needs to be better structured, at the moment it is more of a description of the specialized literature

Author Response

REVIEWER 2

Comment #1:

Title: ...congenital heart disease surgery

ANSWER:

Disease has been added to the title.

Comment #2:

A subchapter explaining the purpose of the study would be welcome

ANSWER:

Section 1.2 Purpose of review (lines 65-74) has been added.

Comment #3:

The Material and Methods chapter is missing, and also the Results section. How was this review done? How many articles were analyzed, what were the inclusion / exclusion criteria? What databases were consulted?

ANSWER:

This paper is a narrative review and as such we cannot provide a strict search algorithm as in systematic reviews. We acknowledge that providing information regarding databases consulted, search terms used, as well as inclusion and exclusion criteria applied, adds clarity to key messages and determines the selection bias. Thus, Material and methods (section 2) and Results (section 3) have been added as per request.

Comment #4:

Your first sentence in the Conclusions chapter talks about " increased cost, morbidity and mortality", but you have not previously discussed all these issues

ANSWER:

Although our study was not designed to look into cost, morbidity and mortality for pediatric patients with CHD developing postoperative chylothorax, all these associations are well described in literature (e.g., ref no 5, 6). We mention this in section 4.1 lines 278-280, without further expanding on the topic, as morbidity and mortality due to chylothorax is not our main points of interest. Despite that, we feel that these associations are based on robust evidence coming out from large databases analyses (PC4 and PHIS). As such, we believe that it is justified to use these data as an introduction to support our conclusions and point out the necessity of our suggestions.

Comment #5:

Your review needs to be better structured, at the moment it is more of a description of the specialized literature

ANSWER:

Structure has been thoroughly revised.

Round 2

Reviewer 2 Report

Dear Authors,

 I find your manuscript much improved. 

I have only two minor remarks to make. 

Lines 488-490: Is this a conclusion of your experience or does it require a reference? 

Table 4 should be moved from the conclusions to the results or discussion section.

Author Response

Dear Reviewer,

Comment #1:

Lines 488-490: Is this a conclusion of your experience or does it require a reference?

Answer: This sentence has been appropriately rephrased.

Comment #2:

Table 4 should be moved from the conclusions to the results or discussion section.

Answer: A short section 4.5 has been added at the end of the discussion part (lines 509-513), which now contains Table 4.
